# Sustainable Development Goals and 2030 Agenda: Awareness, Knowledge and Attitudes in Nine Italian Universities, 2019

**DOI:** 10.3390/ijerph17238968

**Published:** 2020-12-02

**Authors:** Cecilia Smaniotto, Claudio Battistella, Laura Brunelli, Edoardo Ruscio, Antonella Agodi, Francesco Auxilia, Valentina Baccolini, Umberto Gelatti, Anna Odone, Rosa Prato, Stefano Tardivo, Gianluca Voglino, Francesca Valent, Silvio Brusaferro, Federica Balzarini, Martina Barchitta, Alberto Carli, Francesco Castelli, Cristina Coppola, Giuseppina Iannelli, Marica Milazzo, Barbara Rosina, Carla Salerno, Roberta Siliquini, Sauro Sisi

**Affiliations:** 1Department of Medicine, University of Udine, 33100 Udine, Friuli Venezia Giulia, Italy; claudio.battistella@aulss4.veneto.it (C.B.); laura.brunelli@uniud.it (L.B.); ruscio.edoardo@spes.uniud.it (E.R.); silvio.brusaferro@uniud.it (S.B.); 2Friuli Centrale Healthcare and University Integrated Trust, 33100 Udine, Friuli Venezia Giulia, Italy; francesca.valent@asufc.sanita.fvg.it; 3Department of Medical and Surgical Sciences and Advanced Technologies “G.F. Ingrassia”, University of Catania, 95123 Catania, Sicilia, Italy; agodia@unict.it (A.A.); martina.barchitta@unict.it (M.B.); 4Department of Biomedical Sciences for Health, University of Milan, 20122 Milan, Lombardia, Italy; francesco.auxilia@unimi.it; 5Department of Public Health and Infectious Diseases, Sapienza University of Rome, 00185 Rome, Lazio, Italy; valentina.baccolini@uniroma1.it (V.B.); carla.salerno@uniroma1.it (C.S.); 6Department of Medical and Surgical Specialties, Radiological Sciences and Public Health, University of Brescia, 25123 Brescia, Lombardia, Italy; umberto.gelatti@unibs.it; 7Faculty of Medicine and Surgery, University Vita-Salute San Raffaele, 20132 Milan, Lombardia, Italy; odone.anna@hsr.it (A.O.); balzarini.federica@hsr.it (F.B.); 8Department of Medical and Surgical Sciences, University of Foggia, 71122 Foggia, Puglia, Italy; rosa.prato@unifg.it (R.P.); cristina.coppola@unifg.it (C.C.); giuseppina.iannelli@unifg.it (G.I.); marica.milazzo@unifg.it (M.M.); sauro.sisi@unifg.it (S.S.); 9Department of Diagnostic and Public Health, University of Verona, 37134 Verona, Veneto, Italy; stefano.tardivo@univr.it (S.T.); alberto.carli@aulss7.veneto.it (A.C.); 10Department of Public Health Sciences and Pediatrics, University of Turin, 10124 Turin, Piemonte, Italy; gianluca.voglino@unito.it (G.V.); roberta.siliquini@unito.it (R.S.); 11University Research and Documentation Center for the 2030 Sustainable Development Agenda, Department of Infectious and Tropical Diseases, University of Brescia and Brescia Spedali Civili General Hospital, 25123 Brescia, Lombardia, Italy; francesco.castelli@unibs.it; 12Advisory and Career Service, University of Milan, 20122 Milan, Lombardia, Italy; barbara.rosina@unimi.it

**Keywords:** UN Sustainable Development Goals, 2030 Agenda, University, first-year students, sustainability

## Abstract

Sustainable Development Goals (SDGs) and 2030 Agenda represent global development programs. Education can widen the acknowledgement of their relevance and their applications. This survey aims to assess awareness, knowledge and attitudes towards SDGs and sustainability among first-year students in nine Italian Universities. A Likert scale-based online questionnaire of 70 items was compiled by students from March to July 2019. It examined knowledge and expectations referred to sustainable development concepts, indicators and documents/models accounting for sociodemographic variables. Statistical analyses performed were Chi-square test, Fisher’s Exact test, Kendall’s W correlation coefficient, univariate and multivariate analysis. The questionnaire was completed by 1676 students. A low percentage referred a good knowledge of SDGs and 2030 Agenda, most of them had never attended related educational activities previously. Better knowledge of SDGs and 2030 Agenda was observed in case of previous specific educational activities (*p* < 0.001). The expectation towards university guaranteeing an education on SDGs was high, both for personal wisdom and for usefulness in future professional context. A significant difference (*p* < 0.001) in such expectations was found, as healthcare students were less interested than colleagues of other areas. The results showed low knowledge but interest towards sustainable development. A scheduled implementation of academic initiatives should be considered.

## 1. Introduction

### 1.1. Background

The Sustainable Development Goals (SDGs) were adopted in September 2015 by nearly 200 countries participating in a historic UN Summit, widening the scope of the Millennium Development Goals (MDGs), as well as the extent of their ambition, and calling for action each country with equal commitment. The 2030 Agenda, where the SDGs are enounced, represents a globally shared development program, involving the whole population in a common mission aimed to put an end to any form of poverty, to fight against inequalities and to face climate change. The achievement of these goals runs parallel to the guarantee of satisfaction of social needs as quality education, social protection, effective healthcare services and decent work opportunities for all (“No one is left behind”) [1]. The achievement of the specific goal concerning health and well-being in relation to the WHO determinants of health is considered a cornerstone [2], indissociable from the realization of goals to build a healthy society and a healthy planet as well [3,4,5,6,7,8].

In order to achieve these goals of global scope, the majority of the population is expected to know about the SDGs, to recognize their importance and to find application for such goals in any professional context, as well as in personal lifestyle. Nevertheless, the involvement of governments, public and private organizations, as well as academic, scientific, and educational institutions is essential [9]. This commitment must be translated into global partnerships reflecting the multisectoral cooperation and involvement of society at any level, with a specific mention of a required shared course of action on education, health, and gender equity [10]. Expert groups and academic institutions can assume a pivotal role as catalysts in achieving the SDGs, by promoting fight against inequality, definition of lifestyles, healthy work environments and universal healthcare coverage [11]. The ultimate result would be an acceleration in the fulfillment of the 2030 Agenda and in the achievement of its goals [9,12] to promote awareness as well as knowledge and attitudes towards the SDGs in tertiary education [9,12,13]. The establishment of partnerships with universities represents a unique opportunity to give birth to a virtuous positive interaction between creation and dissemination of knowledge, hence also assuring the acquisition of required technical skills to develop and maintain a sustainable society interconnected from an economic, healthcare, social and environmental point of view [14].

The first premise to realize such involvement is represented by adequate competences concerning sustainability themes and the 2030 Agenda’s purposes, and, as every individual can have a role in the implementation strategies to realize the SDGs, an intervention from stakeholder professionals of any field is required, as they are able to elevate sustainable development as a national and supranational priority. As affirmed by UNESCO, the role of a proper education, and more specifically of postsecondary education, for the interpretation and implementation of the SDGs is essential to realize the 2030 Agenda [15]. An essential intersection is thus represented by universities, as a main setting for learning and research before the integration of a professional in the labor market. Universities are historically set to provide and develop subject-specific competences, but they are increasingly asked to cultivate non-specific, generic competences (an integration of knowledge, abilities and attitudes), as indicated by the Bologna Process and the European Higher Education Area [16,17], which aim to promote and implement intergovernmental cooperation between 48 European countries in the field of higher education. They are also an essential context to implement Education for Sustainable Development (ESD), empowering both students and teachers to become vehicle of change in sustainability themes, as already experimented by some national experiences [18,19,20]. Another concept, introduced in 2001 by Sterling [21] is “sustainable education’, which further develops the role of education concerning sustainability themes. Sustainable education is not the mere addition of sustainability concepts to academic curricula, but a cultural shift in the way education and learning is envisioned, with increased ability to face uncertainty and complexity, and an attention in nurturing adaptability, creativity, self-reliance, and resilience.

### 1.2. Literature Review

The acquisition of competences in sustainability, as well as in other fields, can be described through different models, such as Bloom’s Taxonomy of Learning Domains [22], Miller’s Pyramid [23] and the National Center of Education Statistics (NCES) levels [24]. Bloom describes a sequence of domains which progressively lead to competence, starting with simple awareness (that does not necessarily imply interest, meaning being conscious of something) and knowledge (that can be considered a further step, as it implies being familiar with something). Remembering becomes understanding, and then application, analysis, evaluation, and creation follow. Miller and NCES build a different system, considering competence starting with a progression from “knowing” to “knowing how” (knowledge and application of knowledge), and then to “showing how” and “doing” (demonstrating skills and dispositions and implementing them on daily practice). Fuertes and Albareda [25] further explored the acquisition of these competences also as regards sustainability and social responsibility, and how to train and assess them: any competence, as conceived by the aforementioned theories, must, at some point, be put to the test, to “prove” a capacity in performance; the educational context implies a competition element as well. In other words, the academic environment may be the most suitable one not only to guarantee quality in training, but also to create incentives to show proficiency in the acquired skills, achieving a winning combination of knowledge, skills, attitudes, and values.

At least seven dimensions that shape sustainability in higher education have been identified [26]: institutional framework, campus operations, education, research, outreach and collaboration, on-campus experiences and assessment and reporting. A wide literature about experiments in implementing strategies for a more sustainable environment in academic contexts already exists, and explores different dimensions, challenges and issues that must be considered while shaping sustainability-oriented realities in universities, such as awareness, knowledge, engagement, values and beliefs, norms and behaviors or dispositions. Authors from previous studies showed different evidences analyzing these aspects, and sometimes disagree on which of them is the most important in the academic environment. Cogut et al. [27], for example, in a study carried out in University of Michigan collecting data from 2012 to 2015, examined whether awareness could be considered more important than engagement on a long-term effectiveness. The study found the existence of a link between sustainability awareness and increases in sustainable behavior, while engagement was not found to amplify the link between awareness and behavior. More specifically, although other authors [28] found engagement resulting in behavioral change, this American study highlighted how raising students’ awareness determines the acquisition of new ideas and practices, and is overall more effective than simple engagement in dedicated activities in the long term, being awareness more enduring. The study demonstrated how higher levels of operational awareness about sustainability resulted in behavioral change over a four-year period, while engagement increased sustainable behaviors only temporarily. Awareness and knowledge of a topic do not directly change a behavior, but knowledge gain is usually the first step of the path, also according to McGuire’s [29] communication and persuasion matrix, which identifies a sequential nature of making know and understand and then persuade to change, making in this case students willing to change allowing them the time they need to fully understand what sustainability is. Engagement, on the other hand, was found giving a contribution [30], through active learning, in developing a more balanced perspective on sustainability, not only or predominantly including environmental concepts, but social and economic as well. This raises another issue, previously found by Kagawa [31] and further confirmed by other authors [32], as sustainability is mostly perceived by students as a mainly environmental concept, ignoring the scope of sustainability-related themes concerning social and economic elements, and its inherently multi-disciplinary nature. Further analyzing this premise, Lambrechts et al. [33] carried out a segmentation study among business students in 2013–2014, finding four segments in students’ population (moderate problem solvers, pessimistic non-believers, optimistic realists and convinced individualists) but also realizing that there is no universal students’ perspective on sustainability, as students show complex, layered and multi-dimensional attitudes towards sustainability. Another element establishing behavioral changes towards a more sustainable living is represented by normative obligations; the impact of this last element is considerable, as community norms determine long-term changes [34].

The extant literature also offers a group of theories that allow a more detailed analysis of another problem in raising engagement and behavioral change. There is a well-documented inclination to attribute responsibility for a more sustainable living to other people, or organizations. While the term sustainable living is meant as referred to an individual responsibility, according to the displacement theory explained by Chaplin and Wyton [35] there is a discrepancy between declarations from students on their willingness to live sustainably and their actual disposition on adopting consistent behaviors to reach this aim. This is described as a value-action gap, a disengagement that must be addressed, removing those barriers perceived as preventing from acting sustainably, exploring the origin of this displacement of responsibility in internal and external factors, and fixing the existing inverse relationship between perception of importance of a behavior and the ease of participating in that behavior. This low awareness of the impact of individual contributions to sustainability is further highlighted by Eagle et al. [36] and Feygina et al. [37], as a Systems Justification Theory, where lacking in perception of personal responsibility for environmental damage, faulty mental models of causes of environmental, social and economic problems, tolerance of sustainable actions only when they require minimal lifestyle changes (as long as they are not too expensive or demanding), all contribute to the persistence of an idea of maintenance of the status quo and a lack of future orientation, whilst sustainability is not and has never been only about satisfying current needs. Ultimately, the Value-Belief-Norm (VBN) theory states how social and psychological factors, such as values, beliefs and norms, influence behaviors, and finds an ideal application in universities and colleges as students undergo profound changes in their assumptions and identity during their undergraduate years [38] and, unlike most adults, are still in the process of forming their own values and beliefs. A study carried out at the University of Michigan in 2015 [39] surveyed 2828 students on five values considered leading to a more pro-environmental behavior (humanistic altruism, biospheric altruism, egoism, traditionalism, openness to change). The study found strengths in humanistic altruism as well as in biospheric altruism and openness to change, but highlighted that biospheric values were consistently correlated with all the five pro-environmental behavior variables (recycling, sustainable food choices, transportation choices, electricity use and support for pro-environmental candidates). Thus, universities can encourage sustainable behaviors cultivating biospheric or pro-nature values and appealing to students’ beliefs system, while creating marketing campaigns that appeal to different values, rather than taking a “one-size-fits-all” approach. 

Sources of information are important as well, as they contribute as external factors to influence pro-sustainability behaviors up to 80% of an individual awareness [40], including norms, pressures and traditions, family, friends, neighbors, education, media, and social media. Students’ preferred sources and message channels for receiving sustainability information include [32], more than others, posters, professors or instructors, Facebook, friends; television is indicated as a source but less represented than the aforementioned, as well as websites of organizations and family members. The study also highlighted the importance of delivering the message in the way students would like to be, considering from whom and from where, tailoring communication to the target.

In Italy, higher education is offered by universities, high-level arts music and design institutions and higher technical institutions (these latter are an attempt to further expand the non-academic tertiary education, but are still a *niche* phenomenon). They are all autonomous, establishing their own mission and governing bodies, as well as their teaching and research structures. Italy has 95 universities institutions (61 state universities, 17 non-state universities, 6 higher schools and 11 online universities) and 127 higher education institutions for fine arts, music, dance, and design [41].

The quality and effectiveness of the Italian education system is evaluated by the autonomous National Institute for the Educational Evaluation of Instruction and Training (INVALSI), with periodic national assessments every three years. Quality assurance in higher education is carried out with internal institutional evaluations as well as with external evaluations by the National Agency for the Evaluation of the University and Research System (ANVUR). Teachers are assessed after three years of activity to secure their position, through an internal evaluation carried out by the University Internal Evaluation Group. The National Student Registry (ANS) provides data in real time on the academic career of every student enrolled in the Italian university system [42].

Considering the aforementioned premises, the first, essential basis for a proper education on sustainability should be a well-developed investment and support by each country’s government in the educational and training system in general, and in higher education in particular. In Italy, this meets long-lasting difficulties, as shown by several key indicators [41]. Italy invests well below the EU average in education, particularly in higher education. This low investment is, moreover, unevenly spread across educational levels. General government expenditure on education was 3.8% of GDP in 2017, among the lowest in the EU, and is projected to fall over the next 15 years reflecting the country’s demographic decline [43]. Italy has a high level of early school leaving (ESL) when compared to other EU countries (14.5% in 2018 among 18–24 years, while the EU average is 10.6%), and with marked regional differences especially between northern-central regions and southern-islands regions. In 2018, the share of 30–34 year-olds with tertiary educational attainment was 26.9%, the second-lowest in the EU and well below the EU average of 39.9%. Furthermore, highly qualified people have difficulties in finding employment, and the result is that Italian graduates increasingly seek employment abroad [43]. Italy also has to face chronic problems concerning teachers. Among all factors involved in school environment, teachers have a great impact on students’ learning outcomes, and they should be supported to reach high levels of quality and effectiveness in their professional activity. Despite this, Italy has the oldest teaching workforce in the EU, and recent recruitments were ineffective in actually renewing the teaching body: in 2017, 58% in primary and secondary schools were over 50 years old, and 17% were over 60; as regards higher education, over a fifth of the academic staff is over the age of 60, and only 14% is under 40 [42]. The teaching career system offers only a single career pathway with fixed salary increases based solely on seniority, meaning a lack of performance-related incentives. Teachers can reach the maximum salary only after 35 years of service, and schools in disadvantaged areas often are staffed with unexperienced teachers with temporary contracts. Italy does not currently have a national system for teacher appraisal, although from 2015 the *Good School* reform has introduced an annual appraisal carried out by school-based committees. Despite this, Italy has the second-highest proportion of teachers satisfied with their job (96% vs. the EU average of 89.5%) [44].

The primary aim of this study is to assess the overall level of awareness, knowledge and attitudes regarding the SDGs and sustainable development among the new students of the Italian academic community. Secondary aims are: to find out the information sources used in acquiring competences concerning SDGs and sustainable development, as well as specific expectations towards the academic education regarding sustainability; to identify predictors of higher awareness, knowledge and more sustainable attitudes among students; to assess specific knowledge gaps or obstacles to the implementation of sustainable actions.

## 2. Materials and Methods 

### 2.1. Data Collection

The survey was directed at all students who enrolled at university for academic session 2018/2019, for any department for bachelor’s degree. Students who enrolled for master’s degrees, PhDs, specialization course or other post-degree academic paths were excluded from the survey. Students attending the 2nd–6th year, as well as the teaching staff, were not included as well. The decision to consider only first year students came in order to make an assessment of the current level of awareness, knowledge and attitudes on sustainability themes reflecting the background that is currently provided from secondary school, and with the prospect of periodically repeating this assessment during the academic studies, or at the end of the academic path. The possibility of including the teaching staff was considered as well, but it was ultimately reserved for a possible future study.

Students participation was actively promoted among all 82,729 students of nine Italian Universities: Brescia, Catania, Foggia, Milan (Statale and Vita-Salute San Raffaele), Rome, Turin, Udine, Verona; chosen according to convenience criteria as participating in a consolidated Collaborating Group. Participation to the survey was open, anonymous and completely free; collected data were managed with full respect of the existing regulations for privacy (in accordance with the protocol approved by the academic Data Protection Officer of Udine University, n.A102, 21 March 2019) and were used only in aggregated form by the Department of Medical Area of the University of Udine. Participation to the questionnaire available on the EU Survey platform [45] was promoted both via e-mail invitation and banner exhibition on the University websites. Furthermore, additional promotion modalities were implemented as appropriate among direct promotion in classes and during exams; distribution via leaflets with QR Code in University canteens, study halls and at the University Sport Centre; contact with students’ associations for collaboration.

Between 22 March 2019 and 31 July 2019, a structured online questionnaire was administered to first-year Italian university students. 

The questionnaire was composed of 70 items, which were chosen by the research team drawing inspiration from concepts, indicators, documents and models found in the existing literature on SDGs, combining environmental, social and economic elements (considered the three main “areas” of sustainability) [46,47,48,49]. The questionnaire was divided into three sections, which surveyed, respectively: the knowledge on SDGs and sustainability key issues (Part 1); information sources for such topics (Part 2); expectations towards University for the acquisition of such knowledge (Part 3). An additional section collected data about socio-demographic variables: age, gender, Region of provenance, attended University, academic field of current degree course, previous secondary school, previous academic studies, previously attended didactic activities/courses on SDGs or sustainable development. Each section (Part 1–3) assessed eight concepts regarding sustainable development, six indicators and six relevant documents/agreements/models for sustainable development history. Five possible Likert scale-based answers were available for Part 1 (Never heard about it; Heard about it; Talked about it; Studied it at school; Autonomously informed about it) and Part 3 (No, I do not consider it relevant for me; No, I do not think University should teach me about it; It makes no difference for me; Yes, but only for personal wisdom; Yes, also for my professional future); while six different options were given for Part 2 (I have no knowledge concerning this topic; school learning; television; newspapers/magazines/books; web sources e.g., websites, social networks, blogs; other sources). Each item was considered individually for the descriptive analysis related to awareness and knowledge, sources of information and learning attitudes. The 5-point Likert scale for the assessment of awareness/knowledge started with the lowest level (neither awareness nor knowledge of the topic) and progressed with awareness (having heard of the topic), knowledge of the topic (having at least talked about the topic), school study of the topic or autonomous acquisition of knowledge for independent initiative. The assessment of the main sources of information was based on the most common and available sources considering the target population (students, on their first year of academic studies, familiar both with school learning and online, media and family sources) and findings in previous literature [32]. The 5-point Likert scale for the assessment of learning attitudes started with the lowest level (no learning expectations), progressing with no learning expectations in the academic context, neither high nor low learning expectations and then learning expectations for personal knowledge or, on the highest level, also for possible professional usefulness. At the end of the questionnaire completion, links to websites for further information and courses or events organized by each University were made available.

The questionnaire is available online for consultation (link provided at the end of the text in the Appendix A section).

### 2.2. Statistical Analysis

The collection of the questionnaires was carried out in order to reach a convenience sample from the participating Universities. 

Data were analyzed considering the whole national sample and grouping universities according to geographical distribution into two macro-sections: Northern-Italy (Brescia, Milan, Turin, Udine, Verona), and CSI-Italy for Central + Southern + Isles (Catania, Foggia and Rome). Students’ Regions of provenance were grouped in the same way. Such distinction between Northern and CSI Regions was chosen in order to ease the comparison between different geographical provenances, as Northern Universities represented six out of nine Universities. The academic degree courses were clustered into five areas: economic, healthcare, scientific, humanistic areas and other degree. The five possible answers for Part 1 and 3 were dichotomized, in order to consider the first three as negative assumptions (low or superficial knowledge/attitude) and the last two as positive assumptions (good or elevated knowledge/attitude). For the assessment of global knowledge and attitudes, items with positive assumptions were considered on the basis of the total 20 selectable items. To analyze the sources of information, results were elaborated considering every source representation on the 20 surveyed items. Ultimately, a concordance analysis between selected items was performed through Kendall’s W correlation coefficient, in order to evaluate if a declared knowledge or attitude towards a specific item was consistent with the knowledge or attitude towards another pertinent item. Dichotomous and categorized variables were described reporting absolute and relative frequencies. Distribution for assigned scores was described reporting median and range. In order to assess the possible association between categorized variables, Chi-square test and Fisher’s Exact test were applied, as appropriate. Associations in relation to the outcomes of knowledge and attitudes for the item concerning SDGs and 2030 Agenda were further examined applying univariate and multivariate logistic regression analysis. All significant variables resulted by univariate analysis with *p* < 0.10 were included in the multivariate regression model. For every test applied in this study was considered as significant an association with *p* < 0.05. All analyses were performed using Stata/IC 13.0 (StataCorp LP, College Station, TX, USA).

## 3. Results

### 3.1. Socio-Demographic Variables

Overall, 1676 questionnaires were collected, distributed as follows: 234 (14.0%) from Brescia; 258 (15.4%) from Catania; 388 (23.2%) from Foggia; 75 (4.5%) from Milan (Vita-Salute San Raffaele), 29 (1.7%) from Milan (Statale); 158 (9.4%) from Rome; 112 (6.7%) from Turin; 289 (17.2%) from Udine; 133 (7.9%) from Verona. The population of participating students included a prevalence of female subjects (n.1006; 60.0%) with a mean age of 21.05 ± 4.49 years (range 18–71 years). The representation of first-year students according to the enrollments for academic session 2018/2019 in the nine participating Universities was 2.0%. The provenance of the interviewed university students showed a comparable representation of Northern-Italy Regions (n.830, 49.5%) and CSI-Italy (n.846; 50.5%). On a national level, the sample of students showed that one third of them attended a scientific high school before university, followed by those attending technical institutes (16.5%). Other secondary schools were less represented: classical (12.7%), linguistic (7.3%), social pedagogy (6.5%), applied sciences (6.3%), chartered accountant qualification (6.0%), hospitality training institute (4.1%), agricultural engineering qualification (1.6%), artistic (1.0%), primary school teaching qualification (0.4%), musical (0.4%), school of art (0.1%) or other (2.5%). Half of students (49.8%) were currently attending a degree course in the healthcare area, being them medicine, nursing, biotechnologies, sport sciences or healthcare technician degrees. The humanistic area (including literary, juridical, political and linguistic degrees) was represented by 16.9% of respondents; students from economic area contributed for 9.6% of responses and respondents from scientific area (which included engineering, architectural, mathematic and informatics degrees) for 8.2%. Ultimately, 15.5% of respondents declared to attend degree courses not included in the aforementioned categories. Approximately two thirds (71.8%) of students who compiled the questionnaire had not attended other degree course before the current one; when this was the case, mostly it had not been completed (70.4%). Most students (78.1%) reported having never attended any didactic activity or educational initiative specifically related to SDGs or sustainable development. 

The following sub-sections describe the findings on knowledge, sources of information and learning attitudes towards SDGs and 2030 Agenda. It is important, as a preliminary statement, to consider that the evidence collected is limited to the analysis carried out. As the “awareness”/”knowledge” competence was assessed through a Likert scale including both the “awareness” and the “knowledge” level, and was followed by the analysis of the most common sources, the results are presented in two sub-sections: the first one concerning “awareness”/”knowledge” and sources; the second one concerning “learning attitudes”, aspects related to students’ dispositions towards SDGs and 2030 Agenda.

#### 3.1.1. Knowledge and Sources of Information about SDGs and 2030 Agenda

The knowledge of concepts, indicators, documents and models regarding sustainable development was low almost everywhere, with few exceptions. The results obtained on national scale as well as for geographical macro-area concerning the knowledge already gleaned by students about the items surveyed by the questionnaire are reported in Table 1.

Among concepts, a high knowledge resulted only for greenhouse effect, declared to be gained through autonomous information (n.280; 16.7%) or school studies (n.1160; 69.2%) (data not shown). Indicators of sustainability, such as Genuine Progress Indicator, Green Gross Domestic Product or Indicator of Sustainable Economic Welfare (ISEW), are generally little-known, with a moderately better pattern resulted for Human Development Index (HDI). Among documents and models of sustainability, a good knowledge level was found only for more recent international agreements on climate change (Paris Agreement on climate change, 2015) and on the limitation of greenhouse gas emissions (Kyoto Protocol, 1997).

The academic field in which a better knowledge was found was the scientific area (5.0), followed by humanistic area (4.6), other degree (4.6), economic area (4.3) and healthcare area (4.0). A high knowledge concordance was found among Paris Agreement 2015 and Kyoto Protocol 1997 (W = 0.7147). A significant association was found at the univariate analysis between lower knowledge and attendance of degree courses from healthcare area (*p* < 0.001). This association was confirmed by multivariate analysis. Both univariate and multivariate analysis, however, found a positive significant association between the previous attendance of specific non-academic courses on SDGs and a better knowledge (*p* < 0.001). These results are outlined in Table 2.

Nationally, school learning resulted to be the main way for the acquisition of sustainability knowledge (mean of 4.2 items out of 20), while subsequent ones were: online sources (3.6); television (3.4); paper sources such as books, magazines and newspapers (2.3); other sources (0.4).

#### 3.1.2. Learning Attitudes Towards SDGs and 2030 Agenda

The attitudes towards concepts, indicators, documents and models of sustainable development were generally moderate to high, with some exceptions. In particular, learning attitudes for SDGs and 2030 Agenda were moderate, with equal interest in both geographical macro-areas. The highest interest for sustainability concepts was found for greenhouse effect (84.4%), health inequalities (82.2%) and ecological footprint (81.4%). Human Development Index (HDI) and Equitable and sustainable well-being (BES) were the indicators showing highest learning expectations (77.1% and 75.2% respectively). As for documents and models concerning sustainable development, Paris Agreement on climate change and Kyoto Protocol represented the items with highest expectations in this category, respectively 76.0% and 70.9%. The attitudes expressed by students from the two macro-areas were comparable with national results. The learning expectations regarding the surveyed concepts, indicators, documents and models expressed by first-year University students are shown in Table 3.

A positive learning attitude (educational expectations for personal wisdom or for usefulness in their future careers) was found to be higher for the humanistic area (15.4), followed by economic area (15.3), other degree (15.2), healthcare area (13.7) and scientific area (12.9). A high concordance was found between the following learning attitudes: ecological footprint and greenhouse effect (W = 0.7054); ISEW and ecological footprint (W = 0.7164); Paris Agreement 2015, Kyoto Protocol 1997 and Montreal Protocol 1987 (W = 0.7634); Paris Agreement 2015 and Kyoto Protocol 1997 (W = 0.7949); Paris Agreement and Montreal Protocol (W = 0.7606); Kyoto Protocol 1997 and Montreal Protocol 1987 (W = 0.8306). As shown in Table 4, a significant association with lower attitudes was found at the univariate analysis with the attendance of a degree course in healthcare area (*p* = 0.002). On the other hand, a positive significant association (higher learning expectations) was found for female gender (*p* < 0.001) and previous attendance of specific non-academic courses on SDGs (*p* < 0.001). These results were confirmed by multivariate analysis.

## 4. Discussion

The results obtained by our survey are consistent with findings from other research projects, such as the survey created by the European Commission and conducted in December 2015. The results reported in the Special Eurobarometer 441 showed most of the interviewed subjects from European countries (63.0%) having never read nor heard about SDGs, while 26.0%, although having heard about them, reported not knowing what they were; only 36.0% of respondents claimed having heard about them, but only one out of ten actually knew what they were [50]. A specific close-up on students and staff in academic environment is available in literature only for a limited number of contexts. According to a study performed in 2017 within a Nigerian academic community, even if 43.0% of respondents were aware of SDGs, only 4.2% demonstrated a good knowledge level about them (2.4% of students and 9.7% of staff respectively [13]. Another study carried out during 2017 in one of the biggest academic communities in Malaysia [51] outlined a different situation: while most students were found having a moderate (19.5%) to high (74.1%) knowledge concerning sustainability, especially sustainable consumption, they revealed having only a moderate (65.5%) attitude towards a widespread application of such knowledge in their daily life and workplace, mostly because not being used to do it (28.3%). Although these results are limited to the local context where the surveys took place, they are consistent with data collected in the European region and underline the existence of a generally low degree of awareness among both citizens and academic communities, or a lack of the drive to convert sustainability notions into action in everyday life. The findings outlined by the present study confirmed the generally scarce knowledge as regards sustainability themes, except for greenhouse effect and, Kyoto Protocol and Paris Agreement on climate change. This can be the consequence of a more widespread media coverage of these specific issues, as well as, the relatively recent approval of international agreements. Moreover, these specific topics maybe are more easily talked about in didactic programs offered by Italian secondary schools. This finding is consistent with previous studies [31], confirming that sustainability in students’ mind is still mostly perceived as an environmental matter, with little to no consideration for social and economic components. Moreover, similarly with results from other studies [52], a perception of sustainability as a specialized sector, linked to natural sciences, biology or geography, persists, precluding chances to teach sustainability as a multi-disciplinary concept, with connections to any course of degree and studies. As Zsóka et al. [52] found in their study, university students acquire knowledge more independently, but they declare to accept, as favorite sources following their own interest in the topic, education and the media (while friends and acquaintances are at the bottom of their list). Our results confirm this preference, as school learning and online sources, followed by television, were the most represented sources of information declared by the students. As for the moderately higher knowledge of the Human Development Index, this is possibly related to the fact that HDI can be discussed in secondary school dealing with social studies, as well as political history and geography, conversely to what happens for other economic indicators. Lower knowledge and expectations were found for less recent documents, even if still valid (e.g., Brundtland Report, Montreal Protocol) and for sustainably oriented economic models, as Doughnut Economy. Moreover, this latter is linked to the concept of planetary boundaries, for which students showed very little knowledge. Such scarce knowledge might be due to a less detailed information on these themes gleaned from information sources, as well as a lower perception of them as themes of common interest. Furthermore, the attitude of public opinion to single out non-governmental organizations (NGO) and United Nations for main responsibilities in actual attainment of strategies to achieve the SDGs, instead of national governments and multinational corporations [53] should not be forgotten, as described in the aforementioned displacement theory [35], where the majority (75%) of the surveyed students felt that, although there were issues to be addressed concerning sustainability, the solution of these issues was up to someone else, even indicating flatmates and their behavior as barriers for a more sustainable living, clearly struggling to see the practical implications for individual responsibility beyond recognizing the mere importance of the term “sustainability”. Students generally (73% according to Chaplin and Wyton) value sustainability and declare they understand the term; the real challenge is to go forward, as previous findings [32] noticed a rather rigid approach to practical changes in behavior for a more sustainable living, as if it was predominantly an ability to keep going, to maintain the status quo, to preserve, while the overall orientation should be far more active.

The previous attendance of specific non-academic courses on SDGs resulted to be a significant element linked to a better knowledge of SDGs, suggesting that academic study might effectively integrate as educational context for sustainability. Once assured a continuous commitment from universities, such initiative would become a natural process in academic education, as UNESCO [15] and the United Nations [1] indicated.

The significant negative difference found both in knowledge and in attitudes among students from healthcare area deserves to be considered. As for knowledge, it may suggest an even lower previous experience of debate concerning sustainability themes before university. As for attitudes, their low degree might be connected to an erroneous perception of sustainable development and SDGs as subjects without any specific link to healthcare studies and interests.

### Limits and Strengths

The study had to consider some inner limitations as well as some difficulties met while carrying out the survey. 

There were environmental/context issues, as the survey was carried out among students using both online resources and direct administration, though this latter method was mostly carried out during lessons and exam sessions. As regards possible differences between regions of the country, the small number of participants and their distribution among academic and geographical areas may have led to a selection bias, nevertheless by study design we chose to extend invitation to all academic areas and to different Italian regions to limit selection bias as much as possible. 

Then, technical/methodological limitations must be considered. Although the authors limited the number of both items and questions, students may have found difficulties in ending the compilation. Main reasons for survey participation avoidance are unknown (less interest on sustainability?), and a result of that could be an overestimation of both students’ knowledge and attitudes. Despite the high number of items, we chose to administrate the survey online to reduce compiling time and thus permitting us not to lose important information of sustainability themes.

There are some validity and reliability issues, as well, as the questionnaire was developed by a group of Public Health clinicians, drawing inspiration from already existing assessment tools, such as HESI’s SULITEST, but aiming for a simpler tool, focused on a limited (although high) number of questions, with an acceptable time for compilation and avoiding “exam-like” perceptions. Moreover, another limit is represented by the fact that all the answers are self-reported, thus there is no way to verify if what declared by the respondents matches with the real level of the competence.

The questionnaire was developed in order to avoid the perception, for the student, of judgement or assessment, as it was explicitly mentioned that the survey would be completely anonymous and was meant as a contribution to research for academic improvement. Despite this, some students may have been deterred by “exam-like” feelings. 

Ultimately, a possible limitation may have been the conditioning caused by personal beliefs, different cultural background, habits and teachings by the family, investment in the topic. This is a predictable condition, but not modifiable.

There are some strengths for this study that can be summed up. Being the survey completely anonymous, hopefully the respondents felt free to express their knowledge and attitudes. Moreover, to our knowledge this is the first study conducted on university students within the European Region investigating knowledge and attitudes towards SDGs and sustainability themes.

## 5. Conclusions

At the present moment, the acquisition of knowledge and competences regarding sustainable development appears as being left to self-information or post-degree education. This study outlines a framework in which universities could virtuously step in to guarantee and perfect an adequate education of future professionals concerning SDGs, the 2030 Agenda and sustainable development, also following the current increasing popularity and media resonance of these topics. This aim may be achieved by implementing scheduled educational initiatives integrated into the current academic activities, possibly in a nationally coordinated plan. 

The implementation of programmed initiatives to encourage a more sustainable living from university students may focus on different levels, as interventions may increase awareness or knowledge, as well as engagement and sustainable behaviors. This includes considerations for both ESD and sustainable education. The former can be improved enriching competences in sustainability in the teaching body, on several multi-disciplinary aspects, including sustainability in academic courses and exams sessions, workshops, seminars, and conferences. The latter can give students a sustainable model to imitate, using “nudging” techniques and perfecting the existing resources, to promote sustainable behaviors and actions from every subject interfacing with the academic environment. This may include recycling, sustainable food or transportation choices, sustainable infrastructures and energy use, bridging the gender pay gap and promoting gender equality and women’s and minority empowerment, putting an end to any form of discrimination, and establishing education, at any level, as a fundamental human right.

## Figures and Tables

**Table 1 ijerph-17-08968-t001:** Number and percentage of respondents declaring good knowledge for each surveyed item; results reported for Northern and CSI-Italy.

Knowledge about SDGs and 2030 Agenda Related Concepts, Indicators and Documents/Models			
	Northern-Italy	CSI-Italy	Total
	N (%)	N (%)	N (%)
**Concept**			
SDGs and 2030 Agenda	73 (8.4)	54 (6.7)	127 (7.6)
Planetary boundaries	39 (4.5)	64 (8.0)	103 (6.2)
Ecological footprint	283 (32.5)	186 (23.1)	469 (28.0)
Greenhouse effect	754 (86.5)	686 (85.3)	1440 (85.9)
Resilience	282 (32.3)	291 (36.2)	573 (34.2)
Social gradient in health	111 (12.7)	105 (13.1)	216 (12.9)
Health inequalities	318 (36.5)	260 (32.3)	578 (34.5)
Determinants of health	282 (32.3)	181 (22.5)	463 (27.6)
**Indicator**			
Green GDP	85 (9.8)	136 (16.9)	221 (13.2)
Human Development Index (HDI)	259 (29.7)	239 (29.7)	498 (29.7)
Indicator of Sustainable Economic Welfare (ISEW)	80 (9.2)	71 (8.8)	151 (9.0)
Equitable and sustainable well-being (BES)	152 (17.4)	133 (16.5)	285 (17.0)
Genuine Progress Indicator	52 (6.0)	43 (5.3)	95 (5.7)
Gross National Happiness	98 (11.2)	74 (9.2)	172 (10.3)
**Document/Model**			
The Limits to Growth (1972 Report)	40 (4.6)	62 (7.7)	102 (6.1)
Brundtland Report (1987)	21 (2.4)	40 (5.0)	61 (3.6)
Montreal Protocol (1987)	116 (13.3)	94 (11.7)	210 (12.5)
Kyoto Protocol (1997)	443 (50.8)	323 (40.2)	766 (45.7)
Paris Agreement on climate change (2015)	355 (40.7)	244 (30.3)	599 (35.7)
Doughnut Economy	27 (3.1)	48 (6.0)	75 (4.5)

**Table 2 ijerph-17-08968-t002:** Univariate and multivariate analysis for SDGs and 2030 Agenda high knowledge student profile.

Higher Knowledge Student Profile			
Knowledge level			
Median score (range)	0 (0–4)		
Low/superficial n (%)	1549 (92.4)		
Good/elevated n (%)	127 (7.6)		
**Variable at univariate analysis**	**Higher score rate** **n/n (%)**	**OR** **(95%CI)**	***p*-value**
Gender			
Male	54/670 (8.06)	1	-
Female	73/1006 (7.26)	0.893 (0.619–1.287)	0.54
Macroregion of provenance			
Northern Italy	72/830 (8.67)	1	-
CSI—Italy	55/846 (6.50)	0.732 (0.508–1.054)	0.09
University macroregion			
Northern Italy	73/872 (8.37)	1	-
CSI—Italy	54/804 (6.72)	0.788 (0.547–1.136)	0.20
Area of attended degree course			
Economic	24/161 (14.91)	1	-
Scientific	12/138 (8.70)	0.544 (0.261–1.133)	0.10
Healthcare	35/834 (4.20)	0.250 (0.144–0.433)	<0.001
Humanistic	27/283 (9.54)	0.602 (0.335–1.084)	0.09
Other degrees	29/260 (11.15)	0.717 (0.401–1.281)	0.26
Previous attendance of another degree course			
No	92/1203 (7.65)	1	-
Yes	35/473 (7.40)	0.965 (0.644–1.446)	0.86
Previous attendance of specific non-academic courses on SDGs			
No	50/1309 (3.82)	1	-
Yes	77/367 (20.98)	6.686 (4.580–9.759)	<0.001
**Variable at multivariate analysis**		**OR** **(95%CI)**	***p*-value**
Previous attendance of specific non-academic courses on SDGs		6.090 (4.134–8.971)	<0.001
University macroregion (Northern Italy vs. CSI—Italy)		0.817 (0.553–1.207)	0.31
Area of attended degree course			
Economic		1	-
Scientific		0.470 (0.217–1.020)	0.06
Healthcare		0.314 (0.176–0.560)	<0.001
Humanistic		0.761 (0.408–1.417)	0.39
Other degrees		0.691 (0.375–1.273)	0.24

**Table 3 ijerph-17-08968-t003:** Number and percentage of respondents declaring learning expectations for each surveyed item; results reported for Northern and CSI-Italy.

Learning Attitudes on SDGS and 2030 Agenda Related Concepts, Indicators and Documents/Models			
	Northern-Italy	CSI-Italy	Total
	N (%)	N (%)	N (%)
**Concept**			
SDGs and 2030 Agenda	665 (76.3)	615 (76.5)	1280 (76.4)
Planetary boundaries	587 (67.3)	567 (70.5)	1154 (68.9)
Ecological footprint	713 (81.8)	651 (81.0)	1364 (81.4)
Greenhouse effect	726 (83.3)	688 (85.6)	1414 (84.4)
Resilience	608 (69.7)	559 (69.5)	1167 (69.6)
Social gradient in health	625 (71.7)	575 (71.5)	1200 (71.6)
Health inequalities	704 (80.7)	673 (83.7)	1377 (82.2)
Determinants of health	689 (79.0)	659 (82.0)	1348 (80.4)
**Indicator**			
Green GDP	557 (63.9)	557 (69.3)	1114 (66.5)
Human Development Index (HDI)	654 (75.0)	638 (79.4)	1292 (77.1)
Indicator of Sustainable Economic Welfare (ISEW)	605 (69.4)	590 (73.4)	1195 (71.3)
Equitable and sustainable well-being (BES)	648 (74.3)	612 (76.1)	1260 (75.2)
Genuine Progress Indicator	556 (63.8)	540 (67.2)	1096 (65.4)
Gross National Happiness	591 (67.8)	558 (69.4)	1149 (68.6)
**Document/Model**			
The Limits to Growth (1972 Report)	549 (63.0)	525 (65.3)	1074 (64.1)
Brundtland Report (1987)	496 (56.9)	478 (59.5)	974 (58.1)
Montreal Protocol (1987)	549 (63.0)	507 (63.1)	1056 (63.0)
Kyoto Protocol (1997)	621 (71.2)	568 (70.6)	1189 (70.9)
Paris Agreement on climate change (2015)	662 (75.9)	612 (76.1)	1274 (76.0)
Doughnut Economy	534 (61.2)	505 (62.8)	1039 (62.0)

**Table 4 ijerph-17-08968-t004:** Univariate and multivariate analysis for attitudes towards SDGs and 2030 Agenda.

Higher Attitudes Student Profile			
Expectation level			
Median score (range)	3 (0–4)		
Low/superficial n. (%)	396 (23.6)		
Good/elevated n. (%)	1280 (76.4)		
**Variable at univariate analysis**	**Higher score rate** **n/n (%)**	**OR** **(95%CI)**	***p*-value**
Gender			
Male	479/670 (71.49)	1	-
Female	801/1006 (79.62)	1.558 (1.241–1.956)	<0.001
Macroregion of provenance			
Northern Italy	640/830 (77.11)	1	-
Central Italy, Southern-Italy and Italian Isles	640/846 (75.65)	0.922 (0.736–1.156)	0.48
University macroregion			
Northern Italy	665/872 (76.26)	1	-
Central Italy, Southern-Italy and Italian Isles	615/804 (76.49)	1.013 (0.808–1.269)	0.91
Area of attended degree course			
Economic	133/161 (82.61)	1	-
Scientific	107/138(77.54)	0.727 (0.411–1.286)	0.27
Healthcare	593/834 (71.10)	0.518 (0.336–0.800)	0.003
Humanistic	235/283 (83.04)	1.031 (0.618–1.720)	0.91
Other degrees	212/260 (81.54)	0.930 (0.556–1.555)	0.78
Previous attendance of another degree course			
No	905/1203 (75.23)	1	-
Yes	375/473 (79.28)	1.260 (0.973–1.631)	0.08
Previous attendance of specific non-academic courses on SDGs			
No	970/1309 (74.10)	1	-
Yes	310/367 (84.47)	1.901 (1.396–2.587)	<0.001
**Variable at multivariate analysis**		**OR** **(95%CI)**	***p*-value**
Previous attendance of specific non-academic courses on SDGs		1.848 (1.347–2.534)	<0.001
Gender (Female vs. Male)		1.710 (1.349–2.167)	<0.001
Previous attendance of other degree course		1.291 (0.990–1.683)	0.06
University macroregion (Northern Italy vs. CSI—Italy)		1.011 (0.801–1.276)	0.93
Area of attended degree course			
Economic		1	-
Scientific		0.800 (0.447–1.432)	0.45
Healthcare		0.499(0.320–0.779)	0.002
Humanistic		0.994 (0.590–1.676)	0.98
Other degrees		0.908 (0.539–1.528)	0.72

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
