# Peer review of "Sustainable Development Goals and 2030 Agenda: Awareness, Knowledge and Attitudes in Nine Italian Universities, 2019"

_ijerph, 2020, doi:10.3390/ijerph17238968_

Round 1
Reviewer 1 Report
Review for the authors (2nd)
Sustainable Development Goals and 2030 Agenda: A survey on awareness, knowledge and attitudes among Italian first-year University student, 2019
Abstract, topic and introduction
I continue considering that certain data should not be mentioned in the abstract It would not be necessary to advance percentages that will later be explained in detail in the corresponding section.
In the content of the manuscript, the terms "knowledge" and "attitudes" are now better explained (NCES, 2002). In this sense, tutors are advised to consult for including in line 115 the following reference will allow you to illustrate with a quote the meaning of these explanations:
FUERTES, M. T. & ALBAREDA, S. (2014). Evaluation of generic competences in sustainability and university social responsibility. ACCI, 221-242.
Materials and methods
How were the items related to the survey grouped for the analysis of the current level of awareness, knowledge and attitudes?
Results
In line 40 authors say: This survey aims to assess awareness, knowledge and attitudes towards SDGs and sustainability and then in the results:
Line 262 and 263: The following sub-sections describe the findings on knowledge, sources of information and learning attitudes towards SDGs and 2030 Agenda"
Then: which are the objectives of this research in relation to what the authors have mentioned in the introduction?
The limitations section has been improved
Conclusions
Line 401 and 402 This aim may be achieved by implementing scheduled educational initiatives integrated into the current academic activities…
Could the distinction between “sustainable education” and “education for sustainable development” be taken into account when the authors speak about "educational initiatives integrated into the current academic activities?
Author Response
- The percentages later described in the manuscript are now removed from the abstract.
- The suggested reference is now included in the introduction to better explain the differences between the terms “knowledge” and “attitudes” and how they can be used to evaluate competences in sustainability.
- Method to group the items to analyze awareness, knowledge and attitudes: the text now better explains this aspect (lines 287-299).
- Better explanation of the objectives of the study in introduction-results: the results are now presented in two subsections, the first concerning awareness and knowledge, as well as sources, while the second describes the findings concerning learning attitudes (lines 356-361).
- Educational initiatives that may be implemented into the academic activities: the conclusions now include a mention on how sustainable education and education for sustainable development could be integrated in the current academic context.
Reviewer 2 Report
- The title of the paper is too long, contains more than 15 words. The paper’s title should be short and clear.
- The introduction lacks a systematic overview of literature. The conducted research in this field should be discussed.
- Subsection 3.1.2. is too short (lines 290-293). It is recommended to combine subsections.
- The paper is lacking deeper authors’ contribution and scientific insights.
- The conclusions are too short.
Author Response
- The title is now shorter (15 words).
- Improvement of the ovierview of literature in the introduction and scientific insights from other authors: both the introduction and the discussion have been expanded to include previous works from other authors. The references list now includes 53 elements.
- Subsection 3.1.2. is now combined with 3.1.1.
- Improvement of the conclusions section: the conclusions now include a mention on how sustainable education and education for sustainable development could be integrated in the current academic context.
Reviewer 3 Report
Dear authors, I have read your paper with interest. Although an interesting topic and a large sample, the paper needs thorough revisions before taking into account for publication:
1) The paper lacks theoretical grounding and framing of the research in the existing body of knowledge. Many student perception studies have been published, and I miss reference to these previous studies. More specifically, you can check the following papers:
- Chaplin, G., & Wyton, P. (2014). Student engagement with sustainability: Understanding the value–action gap. International Journal of Sustainability in Higher Education.
- Cebrián, G., Pascual, D., & Moraleda, Á. (2019). Perception of sustainability competencies amongst Spanish pre-service secondary school teachers. International Journal of Sustainability in Higher Education.
- Cogut, G., Webster, N. J., Marans, R. W., & Callewaert, J. (2019). Links between sustainability-related awareness and behavior. International Journal of Sustainability in Higher Education.
- Dalvi-Esfahani, M., Alaedini, Z., Nilashi, M., Samad, S., Asadi, S., & Mohammadi, M. (2020). Students’ green information technology behavior: Beliefs and personality traits. Journal of cleaner production, 257, 120406.
- Eagle, L., Low, D., Case, P., & Vandommele, L. (2015). Attitudes of undergraduate business students toward sustainability issues. International Journal of Sustainability in Higher Education.
- Emanuel, R., & Adams, J. N. (2011). College students' perceptions of campus sustainability. International Journal of Sustainability in Higher Education, 12(1), 79-92.
- Lambrechts, W., Ghijsen, P. W. T., Jacques, A., Walravens, H., Van Liedekerke, L., & Van Petegem, P. (2018). Sustainability segmentation of business students: Toward self-regulated development of critical and interpretational competences in a post-truth era. Journal of cleaner production, 202, 561-570.
- Whitley, C. T., Takahashi, B., Zwickle, A., Besley, J. C., & Lertpratchya, A. P. (2018). Sustainability behaviors among college students: An application of the VBN theory. Environmental education research, 24(2), 245-262.
- Zeegers, Y., & Clark, I. F. (2014). Students' perceptions of education for sustainable development. International Journal of Sustainability in Higher Education.
- Zsóka, Á., Szerényi, Z. M., Széchy, A., & Kocsis, T. (2013). Greening due to environmental education? Environmental knowledge, attitudes, consumer behavior and everyday pro-environmental activities of Hungarian high school and university students. Journal of Cleaner Production, 48, 126-138.
2) This lack of reference to the existing body of knowledge is also visible in the reference list. Only 36 sources are mentioned here, many of which are policy documents (UN/OECD/UNESCO). Although I welcome policy documents with regards to the topic (SDGS/Education), you should considerably expand your review of the literature (suggestions in my previous comment might help here as well).
3) The methods for data collection and -analysis are insufficiently explained. How was the questionnaire developed? Using which constructs/measures/scales/validated lists? What was the survey used (include in appendix)?Etc. You should also explain validity and reliability issues of your method.
4) The results of the survey are presented rather descriptive. I would expect more advanced and additional data analysis. For example, is it possible to define clusters based on differences in survey outcomes? What about specific outcomes such as value-action gap as described in the literature?
5) the discussion and conclusion sections are insufficiently developed. In the discussion, more reference towards the existing body of knowledge is needed. How do your results relate to the existing results?
Author Response
- Improvement of theoretical grounding and framing of the research and of the reference list: both the introduction and the discussion have been expanded to include previous works from other authors, with 53 references now in the list.
- Methods for data collection and analysis (development and validation of the questionnaire, inclusion of the survey in Appendix, validity and reliability issues): the questionnaire was conceived as a first assessment of awareness, knowledge and attitudes competences among first year university students, thus it was in this occasion limited to the analyses carried out and described in the results section. In a possible future research based on this first experience analyses could be extended with cluster assessments, value-action gap and other more specific outcomes. The questionnaire was included as a Supplementary Material and not in the Appendix section as it was considered too long (11 pages) to be included in the main text of the manuscript. Validity and reliability issues are now described in the ‘Limits’ paragraph of the Discussion (lines 497-502).
- Improvement of discussion and conclusion: both sections have been expanded; the discussion includes further links to other previous studies and how our results are consistent with them; the conclusion section now includes a mention on how sustainable education and education for sustainable development could be integrated in the current academic context.
Round 2
Reviewer 3 Report
Dear authors,
thank you for revising your paper according to the reviewer comments, you have addresses all issues.
I would recommend to split up section 1 (introduction) into 2 separate sections: section 1) introduction and section 2) literature review
Author Response
- The introduction is now split into 2 sub-sections: 1.1 Backgroung and 1.2. Literature review
- A further check for English grammar/spelling and more concise language was done, correcting some errors.
This manuscript is a resubmission of an earlier submission. The following is a list of the peer review reports and author responses from that submission.
Round 1
Reviewer 1 Report
In this paper, the researchers identify and quantify an important challenge related to communicating the Sustainable Development Goals (SDG). Policies will only be supported if they are understood. Achieving the ambitious SDGs is only possible when people know about them. While this is an interesting and important problem, and the methodology seems straight forward, appropriate and valuable, this paper does not provide enough scholarly contribution to warrant publication for two major reasons 1. A representative population was not sampled 2. No theoretical contribution or any sort of link to existing theory.
First, these authors conducted a survey with a small (nonrepresentative) sample of the population of Italy to support their claim. They have an impressive number of responses to their survey with 1,676 from several universities in Italy and provide interesting findings related to who knows more based on individual social demographics and interests. However, the classic problem that all academics face persists here, we cannot make wide claims about society when sampling our undergraduate students. Convenience sample means that we always sample our undergrads, but this sample is not sufficient to make wider claims about society. Here, I can see why they are selected in that the authors are arguing that even those who are more educated then most do not know much about the SDGs.
Second, the authors fail to contextualize their work with existing theory related to the SDGs and this specific problem of communicating them in order to achieve them. There is a rich and robust literature related to the SDGs, finding literature is not a challenge. Right now, this paper is a cut and dry basic survey and statical analysis without any really surprising findings. This research needs to be more deeply rooted in the existing literature to situate how the authors' findings support or disprove existing theories and literature about the SDGs. This is glaring in the conclusions section. Also, when looking at the references, it is clear the authors are not advancing theory.
Other issues relate to methodology reporting. It is not clear what exactly the students were asked. How did they ask about the SDG indicators specifically? Or “concepts, indicators, documents and models” related to the SDGs? What does this mean exactly? It says that there were 70 items in the survey – that seems like a lot. Authors say they are not sure why more students did not complete the survey – this might be why.
These authors already have a survey that seems fine and an appropriate analysis plan. I would recommend that the authors share this survey in their social networks to get more responses from people in other countries, even undergrads in other countries. While you are waiting for your responses, bolster the theory section of this paper. There are a number of different ways you could go with theory – but I would recommend drawing on literature related to climate change communication, or participatory governance. Relate the SDGs to what motivates people to change and how people support a policy they understand.
Reviewer 2 Report
The review is in the attached document

Reviewer 3 Report
Reviewer Comments
Thank you for the opportunity to review this paper. I think the statistical component of the paper is generally well presented. It is good to know that attempts are already being made to elucidate the implementation of the 5-year-old/nascent SDGs and the 2030 Agenda, at least at the individual level as this paper tries to.
However, the paper needs to strengthen its methodological and theoretical grounding, in my opinion.
1. The conceptual/theoretical background (and referencing) is currently inadequate in my opinion.
a. The description/explanation of the terms like ‘awareness’, ‘knowledge’, ‘attitudes’, etc. in the context of sustainability is not provided in the introduction nor discussed anywhere. In considering knowledge for example, the mode of acquisition of knowledge, be it through 1st order, 2nd order or 3rd order learning1 matters. Against this backdrop, are the authors suggesting that knowledge as mentioned in this paper (as cognitive identification of a concept, an indicator or documents) presumably as in 1st order learning is indicative that the individual has actually acquired what it takes to effect a behavioural change? Interestingly, ESD the ‘education component’ of sustainable development which is usually associated with knowledge, skills and values is not even briefly described.
b. Another point is the lack of description of the status of the Italian universities with regard to their ability to provide the needed capacity to the students. For example, it is stated in Line 249-250: “The generally positive learning attitude ….” What exactly is that since the terms are not described initially? It is also stated in the abstract that “The expectation towards university guaranteeing an education on SDGs was high (76·4%), … ”. Yet, not much theoretical background is given on the educational capacity of the Italian universities to meet the SDG/SD expectations of these students nor is adequately discussed.
2. Methodologically, the authors state “The primary aim of this study is to assess the overall level of knowledge regarding the SDGs and sustainable development among the new students of the Italian academic community”.
a. Why use first year students instead of say, third or final year students to know the level of impact if the focus is on the universities’ capacity to transfer knowledge? Or the authors are assessing the level of knowledge of these first year undergraduate students primarily gained from high school? In that case, I will consider premise of the study as flawed. This is because the launch of the SDGs and 2030 Agenda, based on the timeframe of the study had been less than 4 years old (in Sept 2015), suggesting that very little related information could have been acquired by the students in high school. It is perhaps no wonder therefore that the level of knowledge is generally low in the results.
b. On the other hand, if the capacity impact of university education on students’ acquisition of the knowledge in question was in essence the aim of this study, then the use of 3rd year/final year students as respondents would have been more appropriate; except if thesurvey is being used as a baseline study for another upcoming survey using the same students in 2 or 3 years’ time.
c. How did the authors come up with the concepts, indicators and reports/models? Were they randomly selected or a number of them were assembled and a particular method used to come up with the final list?
d. Interestingly, the authors mention the Millennium Development Goals (MDGs) in line 2 as the precursor of the SDGs and yet that is not considered among the concepts. Although I should give the authors the benefit of the doubt that the MDGs was somehow erroneously considered to be for the people of the South, clarification is needed on why say, ESD or EE; Agenda 21, UNFCCC, the Convention on Biological Diversity (CBD) were also not considered.
e. Under ‘concepts’, why are 3 items out of 8 on “health” while popular sustainability concepts like ESD, Agenda 21 were not considered? Is it because about 50 percent of the respondents are engaged in health related studies?
3. Discussion and Conclusion: I think these aspects should be strengthened a bit more.
4. Line 280: I suppose this abbreviation for non-governmental organizations (ONG) … should be (NGO).
5. Line 391-391: I got a bit curious with the reference below and hence decided to check it Sustainable Development Goals and 2030 Agenda: a survey on awareness, knowledge and attitudes among 391 Italian first-year University students, 2019. https://ec.europa.eu/eusurvey/runner/SustainableDevelopmentGoals2019. This was the message at the website: “This survey has not yet been published or has already been unpublished in the meantime”. Could you clarify this issue?
1 First order learning contributes very little to sustainability, and second order learning contributes to the sustainability process to an extent that learners can change their previous perceptions after reacting to new sustainability challenges. Third order learning involves the learner engaging the cultural and educational systems in a double learning process to transform it in order to be epistemic, creative or transformative (Sterling, 2002: 15-16). Sterling, S., 2002. Sustainable Education: Re-visioning learning and change. Schumacher
Briefings. No. 6. Green Books Ltd, Bristol. You may also consult: Identifying the factors for sustainability learning performance by Paul Ofei-Manu, Robert J. Didham. Journal of Cleaner Production 198 (2018).
Reviewer 4 Report
The theoretical framework and reference does not include the latest bibliography. The construction of the research instruments must be sufficiently documented and justified. It would have been necessary to have information on the process of creating it. In the case of this questionnaire, we can see items of very different order in each of the blocks in an unjustified way. The responses that have been achieved have been very low, perhaps it could have been avoided with greater involvement of the teaching staff. The analysis, due to the construction of the questionnaire they do not offer relevant information, beyond showing some similarity with results from previous studies. The conclusions are weak.